# Absenteeism Costs Due to COVID-19 and Their Predictors in Non-Hospitalized Patients in Sweden: A Poisson Regression Analysis

**DOI:** 10.3390/ijerph20227052

**Published:** 2023-11-10

**Authors:** Marta A. Kisiel, Seika Lee, Helena Janols, Ahmad Faramarzi

**Affiliations:** 1Environmental and Occupational Medicine, Department of Medical Sciences, Uppsala University, 751 05 Uppsala, Sweden; 2Department of Neurobiology, Care Sciences and Society, Primary Care Medicine, Karolinska Institute, 171 77 Stockholm, Sweden; seika.lee@ki.se; 3Department of Medical Sciences, Section of Infectious Diseases, Uppsala University, 751 05 Uppsala, Sweden; helena.janols@medsci.uu.se; 4Department of Health Economics and Management, School of Public Health, Urmia University of Medical Sciences, Urmia 57147-83734, Iran

**Keywords:** absenteeism costs due to COVID-19, loss of productivity due to COVID-19, Sweden

## Abstract

Background: This study aimed to estimate absenteeism costs and identify their predictors in non-hospitalized patients in Sweden. Methods: This cross-sectional study’s data were derived from the longitudinal project conducted at Uppsala University Hospital. The mean absenteeism costs due to COVID-19 were calculated using the human capital approach, and a Poisson regression analysis was employed to determine predictors of these costs. Results: The findings showed that the average absenteeism cost due to COVID-19 was USD 1907.1, compared to USD 919.4 before the pandemic (*p* < 0.001). Notably, the average absenteeism cost for females was significantly higher due to COVID-19 compared to before the pandemic (USD 1973.5 vs. USD 756.3, *p* = 0.001). Patients who had not fully recovered at the 12-month follow-up exhibited significantly higher costs than those without symptoms at that point (USD 3389.7 vs. USD 546.7, *p* < 0.001). The Poisson regression revealed that several socioeconomic factors, including age, marital status, country of birth, educational level, smoking status, BMI, and occupation, along with COVID-19-related factors such as severity at onset, pandemic wave, persistent symptoms at the follow-up, and newly introduced treatment for depression after the infection, were significant predictors of the absenteeism costs. Conclusions: Our study reveals that the mean absenteeism costs due to COVID-19 doubled compared to the year preceding the pandemic. This information is invaluable for decision-makers and contributes to a better understanding of the economic aspects of COVID-19.

## 1. Introduction

The COVID-19 pandemic, caused by the SARS-CoV-2 virus, has emerged as a global health crisis, profoundly impacting all aspects of life, including economic conditions [1]. The economic implications of COVID-19 encompass both direct medical expenses and indirect costs associated with factors such as illness, premature mortality, and absenteeism [2,3]. Regarding the expenses tied to hospitalization, studies conducted in the USA estimated that the median hospital stay length per patient was 5 days, while average costs of hospitalization per patient were USD 3045, with a median cost of USD 12,046 [4,5]. Another report from China indicated that the mean length of hospital stay was 16.26 days, and the mean expenditure for hospitalization of moderate COVID-19 cases amounted to USD 1177. Additionally, each additional year of the patient’s age corresponded to a 0.9% increase in total hospital expense [6].

In Greece, a study determined that direct costs constituted 18.8% of the overall costs, and, during the period from February to May 2020, 80% of all COVID-19-related cases were attributed to healthcare personnel absenteeism [7]. A study in Iran, focusing on patients requiring hospitalization followed by home recovery, found that COVID-19 directly accounted for 32% of the total economic burden, leaving the remaining proportion to indirect costs, primarily driven by absenteeism [8].

Several studies have showed a rise in sick leave duration and frequency during the pandemic in comparison to the pre-pandemic periods [9,10,11]. However, there are still limited studies on absenteeism due to COVID-19. These few existing studies, for example, from Greece, Iran, and Qatar, based on healthcare workers only, have highlighted the elevated and unpredictable rates of workforce absenteeism associated with COVID-19 [7,12,13]. Notably, absenteeism acts as a critical factor, resulting in reduced productivity, increased expenses, overload of other workers, and potential repercussion on the quality of care delivered [14]. Consequently, a comprehensive exploration of absenteeism costs is essential to grasp the complete economic dimensions of COVID-19, particularly in the post-pandemic period, and to offer valuable insights to aid policymakers in economic recovery efforts [15].

In addition, no prior studies have endeavored to estimate absenteeism costs in Sweden. To address this gap, our study aimed to compute absenteeism costs and identify their predictors among non-hospitalized healthcare workers and other occupational groups with laboratory-confirmed COVID-19, spanning from 10 March to 31 December 2020 in Sweden.

## 2. Materials and Methods

### 2.1. Study Design

The data for this cross-sectional study were derived from the longitudinal project titled “COMBAT post-COVID”, which was conducted at Uppsala University Hospital and detailed in other publications [16,17,18]. The project aimed to monitor non-hospitalized patients with COVID-19 who were receiving follow-up care at the outpatient infectious disease department. The study included data collected from all symptomatic adult patients (18 years old or older) diagnosed with COVID-19 at the Department of Infectious Diseases between 10 March 2020 and 31 December 2020. The categorization of pandemic waves in this study was established by the Swedish Board of Health and Welfare (Socialstyrelsen, Stockholm, Sweden) based on the number of active cases: wave-1 spanning from 10 March 2020 to 30 September 2020 and wave-2 spanning from 1 October 2020 to 30 December 2020 [19]. Patients were eligible for inclusion if they possessed a valid diagnostic test result indicating a positive polymerase chain reaction (PCR) test for SARS-CoV-2 by nasopharyngeal swab, thereby confirming their COVID-19 diagnosis.

A survey was administered to the patients within a time span of 45 to 51 weeks, approximately equivalent to 12 months, following their diagnosis of COVID-19. The distribution of questionnaires primarily occurred via email, facilitated by the use of REDCap [20]. In cases where patients lacked an email contact, the questionnaire was dispatched through traditional postal services to their residential addresses, accompanied by a prepaid return envelope. For those who received the email-based survey, up to two reminders were dispatched at intervals of one and two weeks following the initial contact. Conversely, patients who were sent the standard postal questionnaire did not receive additional reminders due to the constraints of the 12-month post-COVID-19 timeframe. The survey was conducted between 10 March 2021 and 15 January 2022. A total of 401 non-hospitalized adult patients with a confirmed diagnosis of COVID-19 responded to the questionnaire, constituting 56% of the eligible participants, as illustrated in Figure 1.

All responders provided digital or written informed consent when filling out the questionnaire. The study was approved by the institutional ethics committee of Uppsala University Hospital (EPN number 2020-05707) and conducted in accordance with the Helsinki Declaration.

### 2.2. Absenteeism Costs

The outcome measure focused on assessing absenteeism costs. These costs were quantified using the number of self-reported sick leave days attributed to COVID-19 and those occurring a year prior to the pandemic (referred to as “pre-COVID-19”, spanning from January 2019 to February 2020). These sick leave days were utilized to compute the respective costs associated with absenteeism or loss of productivity due to COVID-19 and in the period preceding the pandemic.

The computation of these costs was achieved through the application of the human capital approach. This methodology involves assigning a monetary value to the productivity lost due to factors such as illness or premature mortality with the aim of reflecting an individual’s contribution to the overall national productivity [21]. As per this approach, the monetary value of lost productivity resulting from missed workdays is deemed equivalent to an individual’s daily wage multiplied by the number of days absent from work.

The calculation of lost productivity cost attributed to COVID-19 is expressed in US dollars. To determine the wage value for patients who experienced sick leave, the minimum wages based on the statistics from Sweden were employed [22]. Specifically, for patients with sick leave days, the average monthly salary for Sweden in 2020, which amounted to USD 4125, was utilized to estimate the daily wage value. Consequently, the value of a daily wage was computed as USD 138. All reported costs are standardized to constant 2022 US dollars, utilizing purchasing power parities (PPP) for accuracy and comparability.

### 2.3. Other Included Variables

In this study, various relevant predictors of lost productivity costs attributed to COVID-19 were taken into consideration. These predictors encompassed: gender (male or female); age (divided into four groups: up to 30 years old, 31 to 40 years old, 41 to 50 years old, and over 50 years old); marital status (married/in a partnership or single/divorced/widow(er)); country of birth (divided into four groups: born in Sweden with Swedish parents, born in Sweden with one Swedish parent, born in Sweden but parents not born in Sweden, not born in Sweden); highest educational level (categorized as having received at least 3 years of university education or no university education); smoking status (grouped into three categories: current smokers, former smokers, and never smokers); snuff use (yes or no); Body Mass Index, BMI, in kg/m^2^ (based on self-reported weight and length and divided into: underweight: <18.5; normal: 18.5–25.9; overweight: 25.0–29.9; and obese ≥ 30); pre-existing comorbidity (included hypertension, heart disease, hypo/hyperthyroidism, diabetes type 2, lung disease, cancer, depression/anxiety; grouped as no or one or at least two comorbidities); self-reported severity of COVID-19 onset (mild, moderate, or severe); recovery equal to no self-reported persisting symptoms at 12-month follow-up (yes or no); job status (employed/currently working or on parental leave or not employed, including looking for the job, retired, and student); occupation (classified into healthcare workers with patient contact, including medical doctor, nurse, assistant nurse, physiotherapist, psychologist, occupational therapist, or healthcare workers without patient contact, including managers, and other occupations not related to healthcare, including manual workers and managers); working hours/percent of average 40 h week (divided into 81–100%, 50–80%, and <50%); pandemic wave (first or second); and self-reported information on newly introduced treatment for depression/anxiety (including psychological treatment or medication) after contracting COVID-19.

### 2.4. Statistical Analysis

Categorical variables are presented by providing the number and proportion, while continuous variables are presented with the mean and standard deviation (SD). Absenteeism costs attributed to COVID-19 were computed based on variables during and before the COVID-19 pandemic. A *t*-test analysis was employed to examine the differences in expenses after and before the COVID-19 pandemic. Subsequently, Poisson regression was utilized to explore the factors influencing absenteeism costs resulting from COVID-19 [23,24]. The decision to use Poisson regression was primarily driven by the nature of the outcome variable, which involved count data derived from multiplying the number of days by a specific value. Additionally, the dataset exhibited a significant presence of zero values, with 75% of patients in this study demonstrating no costs. This led to a skewed distribution of the dependent variable.

In Poisson regression, the dependent variable (Y) is limited to non-negative integer values, representing counted objects dependent on specific characteristics (x). The probability of Y, often referred to as the “number of events”, follows a Poisson distribution and is given by:(1)$$P(Y;µ)=\frac{e^{-µ}\;{µ}^{y}}{y!}\;(y=0,1,2,\ldots )$$
where µ represents the average number of events in the Poisson distribution. This distribution is commonly employed to model infrequent occurrences within certain time intervals. In the Poisson regression model, the logarithmic function used is ln(μi) = ηi. This allows the relationship function to be expressed in Equations (2) and (3) as follows:
Ln µ_i_ = β_0_ + β_1_X_1i_ + β_2_X_2i_ + … + β_k_X_ki_(2)
µ_i_ = exp (X_i_^T^β) = exp (β_0_ + β_1_X_1i_ + β_2_X_2i_ + … + β_k_X_ki_)(3)
where i is the observation unit i = 1,2,…, n, and β is a vector of coefficients associated with X_i_. A *p*-value < 0.05 was defined as statistically significant. All statistical analyses were performed using Stata 14 (Stata Corp., College Station, TX, USA).

## 3. Results

### 3.1. The Study Patients’ Characteristics

The human capital approach calculated costs for a working population, and 42 study patients that were not employed due to studies (n = 6), retirement (n = 18), job seeking (n = 13), or being on parental leave (n = 5) were excluded from the cost estimation analysis. There were, in total, 359 non-hospitalized patients working while answering the survey at the 12-month follow-up after COVID-19. The sociodemographic and clinical characteristics of all the responders are presented in Table 1. Approximately 73.4 of the study patients were female. The mean age of the patients was 43.46 years, and 81% were born in Sweden. There were 52.3% who had reported persisting symptom(s) one year after contracting COVID-19. Of the responders, 89.3% were employed/in current work. The total number of missed workdays before and during the COVID-19 pandemic was 2332 (mean = 6.7) and 4823 (mean = 13.8), respectively.

### 3.2. The Lost Productivity Cost (USD) Due to Absenteeism

Table 2 presents the mean lost productivity costs attributed to absenteeism due to COVID-19 during both wave-1 and wave-2 of the pandemic, as well as during the year before the pandemic, in Sweden. The average cost of absenteeism was notably higher for females and, to a lesser extent, for males during the pandemic in comparison to the pre-pandemic period. However, the mean absenteeism costs due to COVID-19 did not exhibit a significant difference between genders. Among the study participants, those aged 41 to 50 experienced the highest average cost of sick leave due to COVID-19 in comparison to other age groups. Healthcare workers with patient contact and also other occupations had a significant increase in absenteeism costs at one-year follow-up. Then, working between 80 and 100% of average working hours (40 h/week), having no coexisting comorbidity, and never having used nicotine products such as cigarette and snuff were associated with higher productivity loss due to COVID-19 absenteeism.

Patients without a university education faced significantly higher mean absenteeism costs during the pandemic than they did before. Participants who had not fully recovered at the 12-month follow-up after contracting COVID-19, those who reported severe symptoms at the onset of COVID-19, and those with newly introduced treatment for depression/anxiety after the infection incurred significantly higher absenteeism costs.

### 3.3. The Predictors of the Higher Absenteeism Costs Due to COVID-19

The Poisson regression identified various factors that predict the occurrence of absenteeism costs due to COVID-19 in non-hospitalized patients, Table 3. These factors included socioeconomic variables such as gender, age, marital status, country of birth, educational level, occupation, BMI, and smoking status. Additionally, COVID-19-related variables such as severity of infection at onset, pandemic wave, persistent symptoms at the follow-up, and newly introduced treatment for depression after COVID-19 were significant predictors. For example, absenteeism costs were, on average, 7.5% higher in men than in women.

## 4. Discussion

The primary findings of this study underscore the substantial increase in mean absenteeism caused by COVID-19 within the timeframe of 10 March to 31 December 2020, as it surpassed twofold when contrasted with the year preceding the pandemic for non-hospitalized patients in Sweden. We showed that the total costs of absenteeism due to COVID-19 were USD 665,574. Our study demonstrated that being female, being between 41 and 50 years old, lacking university education, experiencing more severe COVID-19 symptoms at onset, and enduring persisting symptoms at the one-year follow-up were all significantly linked to heightened absenteeism costs due to COVID-19, as compared to the pre-pandemic period. Furthermore, our research identified various socioeconomic factors, including gender, age, marital status, country of birth, educational level, smoking status, BMI, and occupation, as well as COVID-19-related factors such as severity at onset, pandemic wave, persistent symptoms at follow-up, and newly introduced treatment for depression after the infection, as significant predictors of absenteeism costs.

In accordance with our results, healthcare personnel with patient contact reported lower mean sick leave days due to COVID-19 compared to other occupational groups (12.7 vs. 20.7 days). The previous studies on absenteeism rate due to COVID-19 addressing occupational groups were based only on healthcare personnel and are difficult to compare with our findings due to differences in methodology used to assess absenteeism. However, research from Iran conducted between February and September 2020 revealed that the average duration of absenteeism due to COVID-19 was 16 days, accompanied by an average cost of USD 671.4 per patient [12]. Then, a study conducted in Greece between February and May 2020 demonstrated that the average absenteeism duration due to COVID-19 was 25.8 days [7]. To our knowledge, our study is the first that evaluated absenteeism due to COVID-19 in occupational groups other than healthcare occupational groups. The higher duration of sick leave in other occupational groups might be related to healthcare personnel, despite having higher vulnerability to the infection, being more prone to using protection equipment and keeping physical distance, as recommended from the early days of the pandemic [25].

Our study demonstrated that the mean absenteeism costs due to COVID-19, as compared to costs prior to the pandemic, were notably higher among females compared to males. Furthermore, middle-aged patients exhibited higher costs when contrasted with individuals in other age groups. This observation aligns harmoniously with findings from studies conducted in Iran, Qatar, and Greece, which primarily focused on healthcare workers with milder infections [7,12,13]. Likewise, studies conducted prior to the pandemic indicated that women in Sweden tend to experience higher rates of sickness absence compared to men [26]. Additionally, international studies have suggested that women face lower odds of returning to work following various health conditions [27,28].

Furthermore, our findings revealed a positive correlation between age and the likelihood of incurring absenteeism costs due to COVID-19, with patients between 41 and 50 years old incurring 183% greater costs than those under 30 years, aligning with findings from prior studies [8,12]. A study involving hospitalized patients further illustrated that, with each additional year of a patient’s age, total hospital expenses increased by 0.9% [6].

Furthermore, our analysis indicated that study participants without a university education experienced higher absenteeism costs due to COVID-19, as compared to costs prior to the pandemic, than those with a university education. This finding harmonizes with a previous international study where lower education was associated with extended sick leave due to various diagnoses [29,30]. Higher education has been linked to elevated socioemotional status and a propensity for overall better health, increased life expectancy, and higher job satisfaction [31].

In line with a previous study from Iran, we found that increasing age predicted absenteeism costs due to COVID-19 or increased these costs [12]. Similarly to the study from Qatar, we found that COVID-19-related absenteeism increased during the first wave of the pandemic compared to the pre-COVID-19 period. Moreover, the absenteeism rate was even higher in the second wave compared to the first wave [13]. In addition, we showed that not being fully recovered at the follow-up was a factor associated with absenteeism costs. This finding aligns with a study conducted in China; however, this was a study on direct costs of hospitalization, which reported that healthcare costs for severe cases were 9.5 times higher than for non-severe cases [32]. Then, the previous research showed that coexisting depression/anxiety increases the risk of prolonged sickness absence [9,10,11], and it is in line with our study, where we found that newly introduced treatment for depression/anxiety was a predictor of absenteeism costs.

The novelty of our study is that no previous study on absenteeism costs due to COVID-19 has been conducted in Sweden. In addition, no previous existing studies in the area on non-hospitalized patients considered occupations other than those in healthcare. The strength of our study is that we assessed productivity losses both before and during the pandemic and evaluated predictors of absenteeism costs including several clinical and sociodemographic variables. By using the Poisson regression, we could demonstrate which of the variables predict absenteeism costs. This multi-faceted methodology enhanced the depth and accuracy of our findings.

However, this study had several limitations. We faced the challenge of not having access to information about patients’ salary or about whether they worked full time or part time. Additionally, we lacked individual wage data for each study participant. Instead, we employed the minimum wages per person employed in Sweden in 2020 as a proxy for the daily wage value. Additionally, it is important to note that our study cohort consisted of a relatively small number of participants, with the majority being healthcare personnel and female. This composition could potentially limit the generalizability of our findings. Previous research has consistently demonstrated that healthcare personnel are placed in a high-risk category for COVID-19, subsequently heightening their susceptibility to sick leave due to the infection [17,33]. This occupational category skew is largely due to the prioritization of testing for SARS-CoV-2 in the initial months of the pandemic, with healthcare workers being at the forefront in many countries. Furthermore, our cost estimates could potentially be underestimated due to the exclusion of hospitalized patients from our analysis. Consequently, the derived results from our absenteeism cost estimates might not be readily applicable to the entire Swedish population. Then, our study drew on self-reported retrospective data, encompassing factors such as the duration of sick leave, the severity of disease onset, and the persistence of symptoms at the one-year follow-up. It is important to note that this retrospective nature introduces the potential for reporting bias.

## 5. Conclusions

Our study reveals that absenteeism costs due to non-hospitalized COVID-19 can be significant, with an overall increase of more than twofold during the two first waves of the pandemic compared to the pre-pandemic period. Moreover, our findings emphasize the pivotal role of various socioeconomic factors including gender, age, marital status, country of birth, educational level, occupation, smoking status, and BMI, as well as COVID-19-related factors such as severity of COVID-19 at onset, pandemic wave, persistent symptoms at follow-up, and newly introduced treatment for depression after COVID-19, as significant predictors of absenteeism costs. These results not only underscore the significance of accounting for absenteeism costs when assessing the economic ramifications of COVID-19 but also suggest that strategies to mitigate these costs should be prioritized by policymakers.

## Figures and Tables

**Figure 1 ijerph-20-07052-f001:**
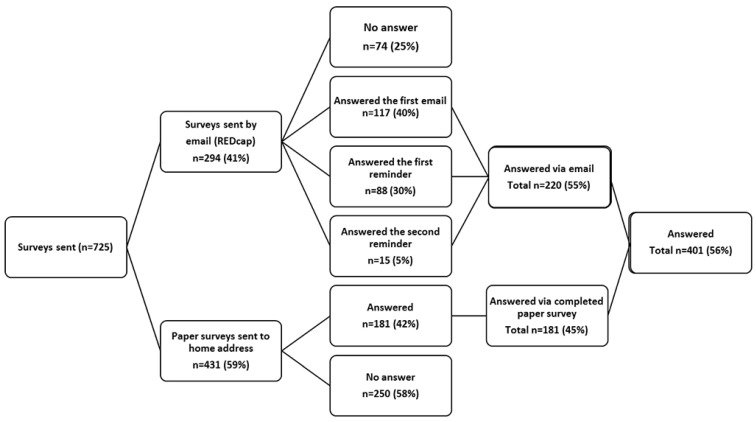
The flow chart of the study population.

**Table 1 ijerph-20-07052-t001:** Characteristics of patients (n = 359) included in the study.

Characteristics	Number of Patients (Percent)
Gender	
Male	93 (26.6)
Female	257 (73.4)
Age (years)	
≤30	67 (19.1)
31–40	83 (23.7)
41–50	87 (24.9)
>50	113 (32.3)
Mean (minimum, maximum)	43.46 (19.69)
Median	44
Marital status	
Married/with partner	238 (68.2)
Single/divorced/widow(er)	111 (31.8)
Country of birth	
Born in Sweden with both parents born in Sweden	243 (70.2)
Born in Sweden with one parent born in Sweden	27 (7.8)
Born in Sweden but parents not born in Sweden	9 (2.6)
Not born in Sweden and neither parent born in Sweden	67 (19.4)
Educational level	
No university	154 (44.1)
University	195 (55.9)
Smoking status	
Ever	83 (23.9)
Never	264 (76.1)
Snuff	
Yes	67 (19.5)
No	276 (80.5)
BMI	
Underweight (<18.5)	5 (1.4)
Normal (18.5–24.9)	180 (51.6)
Overweight (25.0–29.9)	114 (32.7)
Obese (≥30.00)	50 (14.3)
Comorbidity	
No	177 (50.6)
One	106 (30.3)
At least two	67 (19.1)
Self-reported severity of COVID-19 onset	
Mild	147 (42.0)
Moderate	137 (39.1)
Severe	66 (18.9)
Pandemic wave	
First	305 (87.1)
Second	45 (12.9)
Recovered equal to no symptoms at follow-up	
Yes	183 (52.3)
No	167 (47.7)
Newly introduced treatment for depression/anxiety after COVID-19	
Yes	17 (4.9)
No	327 (95.1)
Occupation	
Healthcare with patient contact	237 (68.5)
Healthcare with no patient contact	39 (11.3)
Other	70 (20.2)
Working (%)	
<50	11 (3.1)
50–80	30 (8.6)
81–100	309 (88.3)

**Table 2 ijerph-20-07052-t002:** The lost productivity cost in USD due to absenteeism based on self-reported sick leave days due to COVID-19 and those a year before the pandemic reported by the study patients (n = 359).

Variables	Due to COVID-19	Before Pandemic	*p*-Value ^1^	*p*-Value ^2^
	Mean (SE)	Mean (SE)		
Gender				0.724
Male	1724.3 (609.6)	1370.1 (637.3)	0.295	
Female	1973.5 (364.1)	756.3 (183.9)	0.001	
Age (years)				0.452
≤30	893.9 (435.0)	627.2 (383.5)	0.310	
31–40	1955.3 (702.6)	1191.8 (420.5)	0.325	
41–50	2265.1 (656.3)	510.8 (241.0)	0.016	
>50	2199.4 (590.1)	1207.1 (518.4)	0.018	
Marital status				0.570
Married or with partner	2033.5 (426.9)	1012 (294.1)	0.013	
Single/divorced/widow(er)	1651.0 (358.6)	728.4 (262.3)	0.001	
Country of birth				0.574
Born in Sweden with both parents born in Sweden	1812.2 (374.9)	953.7 (278.2)	0.013	
Born in Sweden with one parent born in Sweden	1640.0 (943.7)	393.6 (201.4)	0.162	
Born in Sweden but parents not born in Sweden	4508.0 (2499.4)	2576.0 (2576.0)	0.070	
Not born in Sweden and neither parent born in Sweden	2133.9 (748.8)	738.2 (370.9)	0.087	
Educational level				0.442
No university	2182.9 (487.3)	1028.7 (378.9)	0.001	
University	1698.0 (407.5)	827.8 (248.6)	0.055	
Smoking status				0.530
Ever	1571.2 (426.6)	1094.0 (451.9)	0.274	
Never	2034.8 (391.7)	874.9 (249.7)	0.001	
Snuff				0.398
Yes	1384.1 (402.7)	896.8 (450.6)	0.368	
No	2062.0 (383.3)	948.2 (252.0)	0.001	
BMI				0.394
Underweight (≥18.4)	0 (0)	0 (0)	-	
Normal (18.5–24.9)	1775.5 (446.0)	745.4 (304.8)	0.008	
Overweight (25–29.9)	1677.8 (416.1)	933.8 (347.8)	0.037	
Obese (≥30.0)	3129.8 (1136.2)	1584.2 (682.7)	0.231	
Comorbidity				0.032
No	1574.8 (387.6)	362.4 (146.8)	0.001	
One	1449.0 (515.9)	598.7 (150.1)	0.115	
At least two	3575.6 (953.0)	2898.0 (1006.0)	0.413	
Pandemic wave				<0.0001
First	1401.3 (214.1)	693.9 (150.9)	0.0005	
Second	5323.7 (1882.0)	2447.2 (1328.2)	0.119	
Self-reported severity at COVID-19 onset				<0.000
Mild	368.0 (94.3)	525.1 (156.9)	0.406	
Moderate	1650.0 (447.9)	1108.4 (433.5)	0.060	
Severe	5929.8 (1249.8)	1405.1 (620.6)	0.001	
Recover = no persistent symptoms at follow-up				<0.0001
Yes	546.7 (141.1)	590.7 (164.4)	0.833	
No	3389.7 (614.8)	1279.5 (415.4)	0.0002	
Newly introduced treatment for depression/anxiety after COVID-19				<0.0001
Yes	9489.5 (3072.9)	5966.5 (3202.0)	0.032	
No	1537.9 (278.7)	670.3 (152.5)	0.004	
Occupation				0.129
Healthcare with patient contact	1756.0 (364.4)	999.0 (235.9)	0.048	
Healthcare workers with no patient contact	569.7 (326.6)	148.6 (75.6)	0.225	
Other	2856.6 (892.3)	938.4 (724.1)	0.003	
Working %				<0.0001
<50	11,240.7 (5725.7)	4127.5 (2214.7)	0.265	
50–80	2350.6 (1612.9)	2898.0 (1771.0)	0.495	
81–100	1530.5 (229.0)	613.1 (152.0)	0.0001	
All included patients	1907.1 (312.2)	919.4 (216.5)	0.0008	-

1: a *t*-test measuring the difference between the productivity costs before the pandemic and during the pandemic due to COVID-19; 2: a *t*-test measuring the difference between the productivity costs during the pandemic due to COVID-19 for each category of variables.

**Table 3 ijerph-20-07052-t003:** Poisson regression results of the absenteeism costs in the study patients (n = 359).

Variables	Estimate	Standard Error	*p*-Value	95% CI	IRR
Gender					
Female	Ref.				
Male	0.073	0.003	<0.0001	(0.067, 0.079)	1.075
Age					
≤30 years	Ref.				
31 to 40 years	0.871	0.005	<0.0001	(0.861, 0.882)	2.390
41 to 50 years	1.042	0.005	<0.0001	(1.032, 1.052)	2.836
>50 years	0.857	0.005	<0.0001	(0.847, 0.867)	2.356
Marital status					
Married or with partner	Ref.				
Single	−0.029	0.003	<0.0001	(−0.035, −0.023)	0.972
Country of birth					
Born in Sweden with both parents born in Sweden	Ref.				
Other	−0.598	0.003	<0.0001	(−0.604, −0.592)	0.550
Educational level					
University	Ref.				
No university	−0.077	0.003	<0.0001	(−0.083, −0.071)	0.926
Smoking status					
Never	Ref.				
Ever	−0.519	0.004	<0.0001	(−0.526, −0.511)	0.595
BMI					
Normal	Ref.				
Other	−0.195	0.003	<0.0001	(−0.201, −0.190)	0.823
Comorbidity					
No	Ref.				
One comorbidity	−0.271	0.004	<0.0001	(−0.278, −0.264)	0.762
At least two	0.008	0.004	0.020	(0.001, 0.015)	1.008
Self-reported severity at COVID-19 onset					
Mild	Ref.				
Moderate	1.102	0.005	<0.0001	(1.092, 1.112)	3.011
Severe	2.023	0.005	<0.0001	(2.013, 2.033)	7.564
Pandemic wave					
First	Ref.				
Second	1.077	0.003	<0.0001	(1.071, 1.083)	2.935
Recovered at 12-month follow−up					
Yes	Ref.				
No	1.405	0.004	<0.0001	(1.397, 1.412)	4.074
Newly introduced treatment for depression/anxiety after COVID-19					
Yes	Ref.				
No	−1.439	0.004	<0.0001	(−1.446, −1.432)	0.237
Occupation					
Healthcare with patient contact	Ref.				
Healthcare with no patient contact	−1.482	0.007	<0.0001	(−1.496, −1.468)	0.227
Other	−0.692	0.003	<0.0001	(−0.698, −0.685)	0.501
Constant	5.237	0.012	<0.0001	(5.213, 5.260)	188.027

CI: confidence interval, IRR: incidence rate ratio.

## Data Availability

Data can be made available on demand.

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
