# Peer review of "Absenteeism Costs Due to COVID-19 and Their Predictors in Non-Hospitalized Patients in Sweden: A Poisson Regression Analysis"

_ijerph, 2023, doi:10.3390/ijerph20227052_

Round 1

Reviewer 1 Report

Comments and Suggestions for Authors

In my opinion, the main contribution of this paper is the detection of variables that predict the absenteeism costs. However, the paper is lacking generalizability. I am wondering if the authors could not use the average wage of the healthcare workers insted of making the aproximation using GDP. I think this should be explained in the paper.

Author Response

Thank you very much for the review comments relating to our manuscript. Thank you again for giving us this opportunity to improve our manuscript.

Replay: We applied the human capital approach to calculated the costs of absenteeism COVID-19. In this method, the estimated costs represent the monetary value of every year of life lost due to mortality or morbidity (in our case due to morbidity), and these costs are calculated from a social perspective. Therefore, in our study, the Gross Domestic Product (GDP) per employed person was used as a proxy for the wage value. GDP per employed person provides a more accurate representation of the value of life compared to the average wage. We addressed this in the limitations section, lines 335-336.

Reviewer 2 Report

Comments and Suggestions for Authors

The research project, aimed to estimate absenteeism costs and assess their predictors in non-hospitalized patients in Sweden linked to COVID-19, was based on a cross sectional survey, derived from a longitudinal project conducted at Uppsala University Hospital. The methodological choices - study type, study population, data collection, cost calculation, statistical analysis, etc. - were appropriate for reaching the set aim and objectives. Despite the limitations, transparently presented by the authors, the results - indicating an overall increase of the absenteeism costs (more than two-fold during two first waves of the compared to the pre-pandemic period), with an emphasis on the pivotal role of variables like age, pandemic wave, disease severity, persistence of symptoms after one year, and treatment for anxiety or depression as significant predictors - are very interesting and may constitute a starting point for further research and health policies addressing COVID-19 pandemic consequences. 

However, I would still have some questions/observations:

 - considering that the commonly accepted active age interval is 15 - 64 y (WHO) and the increasing health risks in the '60s vs. 50's,  why did you choose to include all the subjects with the age > 50 y in the same category? 

 - taking into account subjects' distribution by age (age groups), the minimum, maximum and the median age would also have been significant for the study population description.  

Author Response

Thank you very much for the review comments relating to our manuscript. Thank you again for giving us this opportunity to improve our manuscript

Replay comment 1: Thank you for this comment. We combined the age groups above 50 years (n=64) and above 60 years (n=49) due to their relatively small number of patients. Additionally, the cutoff 50 was used in several previous study on productivity loss due to other diseases. Please see the following references:

  • Eckert KA, Carter MJ, Lansingh VC, Wilson DA, Furtado JM, Frick KD, Resnikoff S. A Simple Method for Estimating the Economic Cost of Productivity Loss Due to Blindness and Moderate to Severe Visual Impairment. Ophthalmic Epidemiol. 2015;22(5):349-55.
  • Stewart WF, Ricci JA, Chee E, Morganstein D. Lost productive work time costs from health conditions in the United States: results from the American Productivity Audit. J Occup Environ Med. 2003 Dec;45(12):1234-46.

Replay comment 2: We addressed this comment, and add required data to the Table 1, page 6.

Reviewer 3 Report

Comments and Suggestions for Authors

Dear authors,

Very interesting and important paper. I have some comments that may help improve the current state of the paper. I have mostly looked at the methodology, as I think it is where I can best review the paper:

1.     In the introduction, several papers from different countries are indicated. I suggest you include the hours/days missed due to absenteeism besides detailing the amount of money related to absenteeism. It is important to know how much absenteeism from COVID-19 changes from country to country, and days/hours is a comparable measure. 

2.     As for the human capital approach, the authors use the GDP per capita to monetize the missing working days. This way of monetization is valid; however, it is a huge overestimation of the costs. Therefore, I suggest using two measures: the minimum hourly wage and the median wage in Sweden. In this way, you have a monetary value of the minimum loss and the median expected loss. This way to estimate missing working hours is closer to the real value.  Alternatively, you can use previously reported salaries of the participants in the survey adjusted for inflation to compute the monetary losses.

3.     I am a bit lost in the statistical methods. You use a GLM model because, among the independent variables, you have several count (non-continuous) variables. You divide the model into two parts, one for those with and without expenses. What kind of expenses do you mean, direct or indirect?

I understand you have the time (days) people worked before COVID and the time (days) people worked after COVID, which can be used to compute the number of missing working days due to COVID-19. Alternatively, you have a variable that directly indicates the missing working days related to COVID-19. I assume this variable has many zeroes, meaning that many people in your sample did not experience any absenteeism, even when diagnosed with COVID-19. If this is the case, you can use a Poisson regression:

Absenteeism_i =  function (age, sex, education, marital status, BMI, smoking, other health conditions, part-time work)

I suggest including an indicator of whether the person works part-time because it may be extremely relevant for absenteeism, i.e., part-time employees may have more flexibility to manage their health condition. 

The predicted values of this regression will give you the E(absenteeism) in your sample, given the specific characteristics included in the equation. These are the estimated working days due to COVID. To monetize these values, multiply them by the minimum hourly wage to get the lower bound of productivity losses. To get a median estimate of the productivity losses, multiply the predicted values by the median wage in Sweden. You can translate these values to yearly losses and compare them to the GDP or health expenditure to see the size of the loss. 

If you want to know which groups lost more, you need to get the median value of the estimated results by age, gender, etc…

Here is a paper with a similar methodology:

Ruoss, M., Brach, M. & Pacheco Barzallo, D. Labor market costs for long-term family caregivers: the situation of caregivers of persons with spinal cord injury in Switzerland.BMC Health Serv Res 23, 676 (2023). https://doi.org/10.1186/s12913-023-09565-7

Author Response

Thank you very much for the review comments relating to our manuscript. Thank you again for giving us this opportunity to improve our manuscript. We provide replays point-by-point. 

In the introduction, several papers from different countries are indicated. I suggest you include the hours/days missed due to absenteeism besides detailing the amount of money related to absenteeism. It is important to know how much absenteeism from COVID-19 changes from country to country, and days/hours is a comparable measure. 

Replay: Thank you for thee comment. We added information on the length of hospital stay in the introduction section, lines 44-46.

As for the human capital approach, the authors use the GDP per capita to monetize the missing working days. This way of monetization is valid; however, it is a huge overestimation of the costs. Therefore, I suggest using two measures: the minimum hourly wage and the median wage in Sweden. In this way, you have a monetary value of the minimum loss and the median expected loss. This way to estimate missing working hours is closer to the real value.  Alternatively, you can use previously reported salaries of the participants in the survey adjusted for inflation to compute the monetary losses.

Replay: Thank you for this comment. Unfortunately, we do not have information about salary of the patients. We addressed this in line 330-331, in the limitations section. We applied the human capital approach to calculated the costs of absenteeism COVID-19. In this method, the estimated costs represent the monetary value of every year of life lost due to mortality or morbidity (in our case due to morbidity), and these costs are calculated from a social perspective. Therefore, in our study, the Gross Domestic Product (GDP) per employed person was used as a proxy for the wage value. GDP per employed person provides a more accurate representation of the value of life compared to the average wage. We addressed this in the limitations section, lines 335-336.

I am a bit lost in the statistical methods. You use a GLM model because, among the independent variables, you have several count (non-continuous) variables. You divide the model into two parts, one for those with and without expenses. What kind of expenses do you mean, direct or indirect?

Replay: Thank you for this comment. The costs of absenteeism are considered a part of indirect costs. We indicated this information in the manuscript, lines 111-112.

I understand you have the time (days) people worked before COVID and the time (days) people worked after COVID, which can be used to compute the number of missing working days due to COVID-19. Alternatively, you have a variable that directly indicates the missing working days related to COVID-19. I assume this variable has many zeroes, meaning that many people in your sample did not experience any absenteeism, even when diagnosed with COVID-19. If this is the case, you can use a Poisson regression: Absenteeism_i =  function (age, sex, education, marital status, BMI, smoking, other health conditions, part-time work)

Replay: Thank you for the comments. The two-part model consists of a Logit or Poisson regression for the first part, which assesses costs between zero and non-zero, and a Generalized Linear Model (GLM) for the second part that assesses costs for non-zero values. We opted for Logit over Poisson regression for the first part because Logit regression allows for the estimation of odds ratios, which are more beneficial for our studies. We addressed this in lines 325-326.

I suggest including an indicator of whether the person works part-time because it may be extremely relevant for absenteeism, i.e., part-time employees may have more flexibility to manage their health condition. 

Replay: Thank you for this comment. We do not have information on weather patients worked full-time or part-time. We added this to the limitation section, lines 336-337.

The predicted values of this regression will give you the E(absenteeism) in your sample, given the specific characteristics included in the equation. These are the estimated working days due to COVID. To monetize these values, multiply them by the minimum hourly wage to get the lower bound of productivity losses. To get a median estimate of the productivity losses, multiply the predicted values by the median wage in Sweden. You can translate these values to yearly losses and compare them to the GDP or health expenditure to see the size of the loss. If you want to know which groups lost more, you need to get the median value of the estimated results by age, gender, etc…Here is a paper with a similar methodology:

Ruoss, M., Brach, M. & Pacheco Barzallo, D. Labor market costs for long-term family caregivers: the situation of caregivers of persons with spinal cord injury in Switzerland.BMC Health Serv Res 23, 676 (2023). https://doi.org/10.1186/s12913-023-09565-7

Replay: We are grateful for your suggestion. Indeed, we utilized the human capital approach to estimate absenteeism costs from a social perspective. However, we do lack information regarding whether patients were employed full-time or part-time, lines 335-337. Nonetheless, your suggestion could serve as a subject for future studies.

Round 2

Reviewer 3 Report

Comments and Suggestions for Authors

Dear authors,

Thank you for the revision and for responding to my comments. I still think two points need to be addressed:

1. Using the GDP per capita is a huge overestimation of the monetary losses due to absenteeism. Instead, a closer estimation is the median salary in Sweden (at the country level). You can compare the level of overestimation just by comparing the two measures. The human capital approach aims to estimate the losses of reduced health status. To make the estimation valid, monetizing such costs should be closer to reality. 

2. Using the Poisson model is more accurate for your data type. You can estimate the marginal effects or odds ratio as well from a Poisson. 

Author Response

Dear Academic Editors,

Thank you again for the review comments relating to our manuscript.

 We have carefully read and taken the concerns into consideration. The responses to the reviewers’ comments are listed below. A revised manuscript file with tracked changes ave been uploaded below as word-file. The line numbers refer to the tracked version.

 Thank you again for giving us this opportunity to improve our manuscript.

Review3: Using the GDP per capita is a huge overestimation of the monetary losses due to absenteeism. Instead, a closer estimation is the median salary in Sweden (at the country level). You can compare the level of overestimation just by comparing the two measures. The human capital approach aims to estimate the losses of reduced health status. To make the estimation valid, monetizing such costs should be closer to reality. 

Replay: Thank you for this comment. As suggested, we changed to the minimum wages based on the statistics Sweden. Specifically, for patients with sick leave days, the average monthly salary for Sweden in 2020, which amounted to $4,125, was utilized to estimate the daily wage value. Consequently, the value of a daily wage was computed as $138. Line 167-170. Then, we changed in the results accordingly, table 2.

Review3: Using the Poisson model is more accurate for your data type. You can estimate the marginal effects or odds ratio as well from a Poisson. 

Replay: Thank you for the comment. As suggested, we changed the analysis to the Poisson model. We changed accordingly in the methods (line 206-333), re-calculated the results (table 3) and revised the discussion (line 578-582).

Replay: Thank you for the comments.

Best Regards,

Marta Kisiel and Ahmad Faramarzi on behalf of other co-authors

2023-11-05
